# Comprehensive Profiling of Alternative Splicing and Alternative Polyadenylation during Fruit Ripening in Watermelon (*Citrullus lanatus*)

**DOI:** 10.3390/ijms242015333

**Published:** 2023-10-18

**Authors:** Yongtao Yu, Yuxiang Liufu, Yi Ren, Jie Zhang, Maoying Li, Shouwei Tian, Jinfang Wang, Shengjin Liao, Guoyi Gong, Haiying Zhang, Shaogui Guo

**Affiliations:** 1National Watermelon and Melon Improvement Center, Beijing Academy of Agriculture and Forestry Sciences, Key Laboratory of Biology and Genetic Improvement of Horticultural Crops (North China), National Engineering Research Center for Vegetables, Beijing Key Laboratory of Vegetable Germplasm Improvement, Beijing 100097, Chinawangjinfang@nercv.org (J.W.);; 2College of Forestry, Basic Forestry and Proteomics Research Center, Fujian Agriculture and Forestry University, Fuzhou 350002, China

**Keywords:** watermelon, alternative splicing, alternative polyadenylation, lncRNA, fruit ripening

## Abstract

Fruit ripening is a highly complicated process that is accompanied by the formation of fruit quality. In recent years, a series of studies have demonstrated post-transcriptional control play important roles in fruit ripening and fruit quality formation. Till now, the post-transcriptional mechanisms for watermelon fruit ripening have not been comprehensively studied. In this study, we conducted PacBio single-molecule long-read sequencing to identify genome-wide alternative splicing (AS), alternative polyadenylation (APA) and long non-coding RNAs (lncRNAs) in watermelon fruit. In total, 6,921,295 error-corrected and mapped full-length non-chimeric (FLNC) reads were obtained. Notably, more than 42,285 distinct splicing isoforms were derived from 5,891,183 intron-containing full-length FLNC reads, including a large number of AS events associated with fruit ripening. In addition, we characterized 21,506 polyadenylation sites from 11,611 genes, 8703 of which have APA sites. Kyoto Encyclopedia of Genes and Genomes (KEGG) analysis showed that fructose and mannose metabolism, starch and sucrose metabolism and carotenoid biosynthesis were both enriched in genes undergoing AS and APA. These results suggest that post-transcriptional regulation might potentially have a key role in regulation of fruit ripening in watermelon. Taken together, our comprehensive PacBio long-read sequencing results offer a valuable resource for watermelon research, and provide new insights into the molecular mechanisms underlying the complex regulatory networks of watermelon fruit ripening.

## 1. Introduction

Watermelon (*Citrullus lanatus*, 2 n = 2 × = 22), a typical non-climacteric fruit, originated in Africa, belongs to the xerophytic genus *Citrullus* Schrad. ex Eckl. et Zeyh. of the botanical family *Cucurbitaceae* and is an important cucurbit crop grown throughout the world. Watermelon is an important experimental model to study the molecular mechanisms of non-climacteric fruits’ ripening [1]. This reflects its economic value and many favorable genetic characteristics, such as relatively small genome size (359.8 Mb), diploid cultivar, short life cycle (3–4 months), and efficient genetic transformation system [2].

Ripening of fleshy fruits is a highly complex process that involves dramatic changes in sugar content, fruit color, fruit texture, flavor, and aroma [3,4]. In recent years, the mechanisms of fruit ripening have been studied extensively at the transcriptional level, and many transcription factors as well as 5-methylcytosine (5mC) DNA methylation at various gene loci have been identified and proven to be responsible for transcriptional regulation of gene expression during the ripening process [5,6]. In addition, a series of studies have demonstrated that posttranscriptional control also plays vital roles in fruit ripening such as AS, APA, mRNA N6-methyladenosine RNA modification methylation and noncoding RNAs [7,8,9].

AS and APA are two important post-transcriptional regulatory mechanisms that increase transcriptomic and proteomic diversity in eukaryotic organisms [10,11]. AS is the process through which multiple transcript isoforms are produced from a single gene [12]. Previous short-read RNA-seq analysis has shown that up to 40–60% of intron-containing genes are alternatively spliced in different tissues or developmental stages, suggesting AS plays key roles in the regulation of development in the model plants, such as *Arabidopsis thaliana* and rice [13,14,15,16]. APA is a widespread phenomenon, generating mRNAs with distinct 3′ ends [11]. The large-scale studies using transcriptome-wide techniques have revealed that 70% of *Arabidopsis* genes and around 50% of rice genes have at least one alternative poly(A) site [17,18]. Single-molecule real-time sequencing (SMRT), developed by Pacific BioSciences (PacBio), offers longer read lengths than the RNA-seq which are needed for transcriptome assembly, making it well-suited for unsolved problems in transcriptome research [19].

Over the past decade, single molecule sequencing technology has been used to characterize the transcriptome diversity in *Salvia miltiorrhiza* [20], *Z. mays* [21], *Sorghum bicolor* [22], *Fragaria vesca* [23], *Phyllostachys edulis* [24], *Trifolium pratense* L. [25], *Ricinus communis* [26], *Ginkgo biloba* [27], *Populus* [28], *Solanum betaceum Cav* [29] and so on. In these studies, most of the full-length cDNAs were generated by combining in equal amounts total RNA from different tissues and/or did not make replicates. In this study, we firstly carried out SMRT long-read sequencing of four critical developmental stages in watermelon (three repeats for each stage). A total of 7,837,450 reads of inserts (ROIs) were generated for 12 samples. Finally, 6,921,295 error-corrected and mapped FLNC reads were obtained. The inclusion of more high-quality FLNC reads is more conducive to conduct AS and APA analyses, which can better reflect the dynamic changes during fruit ripening at post-transcriptional level.

## 2. Results

### 2.1. Watermelon Fruit Transcriptome Sequencing Using PacBio Iso-Seq

Samples were collected at four key developmental stages (10 days after pollination (DAP), 18 DAP, 26 DAP and 34 DAP) (Figure 1), and each stage had three repeats. These samples were the same as in our previously published study [30]. In this current study, PacBio sequel II platform was used to generate long reads for the above 12 samples. A total of 7,837,450 reads of inserts (ROIs) were generated for 12 samples, and on average 88% of them were full-length non-chimeric (FLNC) reads with the entire transcript region from the 5′ to the 3′ end based on the inclusion of barcoded primers and the 3′ poly(A) tails. A total of 843,963,605 paired-end (PE) reads from our previously published transcriptomic data were used to correct the FLNC reads, then the corrected FLNC reads were mapped to the watermelon genome (97103, V2.5 version) using minimap2 [31]. Finally, we obtained 6,921,295 error-corrected and mapped (high-quality) FLNC reads, and the average FLNC reads of 12 samples is 576,774 (Table 1). These FLNC reads cover 73.49% of annotated genes, and 14,520 genes were supported by at least three PacBio reads (Appendix A). The repeatability analysis of 12 samples showed that the results of APA quantification were reliable (Appendix A).

### 2.2. Identification of Full-Length Splicing Isoforms

Till now, the study of the watermelon transcriptome is not totally complete, in particular the annotation of full-length splice isoforms. We obtained a total of 5,891,183 intron-containing FLNC reads that collapsed into 42,285 distinct full-length splice isoforms representing 135,772 unique introns. The consensus donor sites and acceptor sites of watermelon are TTAAAG|GTAAG and TGCAG|GTTAAA, respectively (Figure 2A). The consensus donor and acceptor sites were similar to those in rice and moso bamboo [24,32]. The median number of introns in intron-containing genes in Arabidopsis [33] and moso bamboo [24] is four and seven, respectively, whereas that for each unique splicing isoform in watermelon is six. Due to the limit of read length, it is challenging to identify full-length splicing isoforms accurately by RNA-seq, while PacBio long reads could produce more comprehensive annotation of splicing isoforms with superior contiguity [24,34]. Therefore, we examined the full-length transcript structures of distinct splicing isoforms using PacBio long-read sequencing data. In total, 16,495 AS events were identified without transcriptome assembly, so any possible artificial result could be avoided (Appendix A). We further classified these AS events into seven distinct types: 5574 retained introns (RI), 3557 skipping exon (SE), 3104 alternative 5′ splice-site (A5), 3141 alternative 3′ splice-site (A3), 697 alternative first exon (AF), 267 alternative last exon (AL) events and 155 mutually exclusive exons (MX) (Appendix A). Consistent with previous studies, intron retention comprised the majority of AS events [24,32]. KEGG analysis of genes undergoing AS showed that fructose and mannose metabolism, starch and sucrose metabolism and carotenoid biosynthesis were enriched, which included *tonoplast sugar transporter 2* (*TST2*), *Sugars Will Eventually Be Exported Transporter 3* (*SWEET3*), *insoluble acid invertase* (*IAI*, *Cla97C05G099220*), *UDP-Gal/Glc PPase* (*UGGP*, *Cla97C08G148810*), *Phytoene synthase* (*PSY*, *Cla97C01G008760*) and *Phytoene desaturase* (*PDS*, *Cla97C07G142100*) (Appendix A). All the enriched KEGG terms are listed in Appendix A. Next, we validated the retained intron event of *PDS* (*Cla97C02G037750*, carotenoid biosynthetic enzyme) by RT-PCR. Consistently, RT-PCR could amplify two bands, which also meet the expected size (Figure 2B,C). Considering the relatively low sequencing depth of PacBio long-read sequencing, we quantified differential splicing events using Illumina short reads. The results are shown in Appendix A.

### 2.3. Profiling of Global APA Sites and Differential APA

Global establishment of the accurate polyadenylation cleavage sites to define the 3′ end of genes will greatly improve the gene annotation. PacBio long-read sequencing produces full-length of cDNA, which is suitable to identify APA events. In this study, we firstly carried out long-read sequencing to study the global polyadenylation events at single-nucleotide resolution in watermelon. In total, we characterized 21,506 polyadenylation sites from 11,611 genes, and 8703 genes contained two or more polyadenylation sites (Figure 3A, Appendix A). Furthermore, a total of 4959 intronic polyadenylation sites were identified (Appendix A). Interestingly, these introns were longer than those without polyadenylation sites (Figure 3B). Next, we validated the APA event of *Cla97C01G008870* (Isocitrate dehydrogenase (ICDH), citric acid biosynthetic enzyme) by 3′-RACE. Consistently, PCR could amplify multiple bands, which represented different 3′ ends for transcript variants (Appendix A). Motifs around the watermelon polyadenylation region contained UGUA and AAUAAA (Appendix A), which were similar to previous studies in *Sorghum bicolor*, *Ginkgo biloba* and moso bamboo [22,24,27]. Proximal and distal poly(A) cleavage sites displayed extremely similar distribution, suggesting that proximal sites are real poly(A) cleavage sites (Figure 3D–E). Similar to AS, KEGG analysis showed that fructose and mannose metabolism, starch and sucrose metabolism and carotenoid biosynthesis were also enriched in genes undergoing APA (Figure 3C, Appendix A), which included *IAI* (*Cla97C05G099220*), *Sucrose Synthase* (*SUS*, *Cla97C10G194010*), *Trehalose 6-phosphate phosphatase* (*TPP*, *Cla97C07G136350*, *Cla97C01G018360*), *PSY* (*Cla97C01G008760*, *Cla97C07G137500*), *PDS* (*Cla97C07G142100*, *Cla97C10G198770*) and *9-cis-epoxy-carotenoid dioxygenase* (*NCED*, *Cla97C07G137260*, *Cla97C01G024630*). All the enriched KEGG terms are listed in Appendix A. Finally, we quantified differential APA, and the results are shown in Appendix A.

### 2.4. Identification of Long Non-Coding RNA from PacBio Sequences

In plants, lncRNAs are largely involved in regulation of plant growth and development, synthesizing secondary metabolites, and responding to environmental stress [35]. Recently, a series of noncoding RNAs (ncRNAs), especially lncRNAs, have been functionally characterized in fruit quality formation and fruit ripening at the posttranscriptional level [7]. Using PacBio long-read data, we identified a total of 594 high-confidence lncRNAs in watermelon fruit, which presented a shorter exon length than that of coding genes (Figure 4A, Appendix A). Furthermore, 72 lncRNAs were up-regulated and 79 lncRNAs were down-regulated in 10 DAP vs. 18 DAP group, 37 lncRNAs were up-regulated and 12 lncRNAs were down-regulated in 18 DAP vs. 26 DAP group, and 26 lncRNAs were up-regulated and 14 lncRNAs were down-regulated in 26 DAP vs. 34 DAP group (Figure 4B–D, Appendix A). These differentially expressed lncRNAs might play important roles in watermelon fruit ripening.

## 3. Discussion

Watermelon is an important fruit crop, and an ideal model plant for studies on non-climacteric fruit ripening. In this study, using PacBio single-molecule long-read sequencing, we present the full-length transcriptome of four key developmental stages in watermelon. In total, we obtained 6,921,295 error-corrected and mapped FLNC reads. In addition, 843,963,605 paired-end (PE) reads from our previously published study were also used in this work. These high-quality datasets provided a solid foundation for our subsequent analyses.

AS and APA play a crucial role in the functional diversity of a gene by generating different isoforms to regulate gene expression at the post-transcriptional level in higher eukaryotes. To date, genome-wide analyses of alternative splicing have been studied in some fruits, including tomato [36], kiwifruit [37], olive (Olea europaea) [38], papaya [39], peach [39], melon [39], cucumber [39], blueberry [40], citrus [41], grape (*Vitis vinifera*) [42], sweet cherry (*Prunus avium*) [43], strawberry [23,44] and watermelon (*Citrullus lanatus*) [45]. All the studies conducted RNA-seq to identify AS events except in strawberry and olive, which employed PacBio long-read sequencing. In cucumber and melon, AS was significantly more prevalent at the ripe stage than at the immature stage, while the opposite trend was shown in papaya and peach, implying that developmental stages adopt different alternative splicing strategies for their specific functions [39]. In addition, all vegetative tissues of tetraploid watermelon except the fruit displayed an increased level of AS than diploid watermelon throughout the growth period [45]. In this study, we found AS events were significantly increased at the ripe stage (26 DAP and 34 DAP) than at the immature stage (Figure 2D), which was similar to cucumber and melon.

Common RNA-seq could not provide a better understanding of genome-wide profiling of polyadenylation sites [46]. Previous developed polyadenylation site sequencing (PAS-seq) was used to comprehensively identify plant APA events [17,47]. However, the method has an internal priming issue in that the oligo (dT) primer could also bind internal A-rich sequences [48]. PacBio long-read sequencing generated full-length structures of each splicing isoform and allowed the identification of exact polyadenylation sites [22]. Till now, global APA identification during fleshy fruit development has rarely been reported. In this study, PacBio sequencing was employed to generate long reference sequences and profiling of APA from 8703 genes in watermelon. KEGG analysis showed that fructose and mannose metabolism, starch and sucrose metabolism and carotenoid biosynthesis were enriched in genes undergoing AS and APA, respectively. Interestingly, *IAI* (*Cla97C05G099220*), (*PSY*, *Cla97C01G008760*) and *PDS* (*Cla97C07G142100*) were simultaneously regulated by both AS and APA. Taken together, these results suggested that AS and APA might play important roles in watermelon fruit ripening. In total, 4959 introns from 3073 genes included intronic polyadenylation, which might encode shorter proteins (Ozso-lak et al., 2010). In Arabidopsis and moso bamboo, introns with polyadenylation sites always contain transposable elements (TEs), which result in introns with polyadenylation sites are longer than that without polyadenylation sites [24,49,50]. In addition, previous studies showed that association of intronic TE elements through epigenetic regulation (DNA methylation, histone modification) impact the intronic polyadenylation in Arabidopsis [49,50]. In this study, we also found that introns with polyadenylation sites are longer in watermelon fruit (Figure 3B). In future, more fruit ripening-related epigenetic data will reveal the interplay between epigenetic regulation and intronic polyadenylation in watermelon, such as histone modification and DNA methylation.

Based on our previously published transcriptomic data, we found that a few splicing-related factors displayed obvious up-regulated expression during watermelon fruit ripening such as serine/arginine-rich splicing factor (SR), DEAD-box ATP-dependent RNA helicase (DDX), cyclin-dependent kinase g-2 (CDK), and pre-mRNA-processing factor 39 (PRP39) (Figure 5). Additionally, APA-associated factors also showed different expression during ripening such as cleavage and polyadenylation specificity factor (CPSF), poly(A) polymerase (PAP) and pre-mRNA cleavage factor Im 25 kDa subunit 1 (CFIm25) (Figure 5). These differential splicing-related and APA-associated genes might take part in regulation of AS and APA during fruit ripening in watermelon.

Climate change will have profound effects on food production. The production and quality of fresh fruit can be directly or indirectly affected by exposure to high temperatures and elevated levels of carbon dioxide and ozone. Accelerated crop improvement (traditional breeding methods, genetic modification and gene editing), and new cultivation and processing techniques are required to produce sufficient food supplies and/or economize the production cost, so as to meet the basic nutrient requirements of the growing human population in this age of climate change. Our comprehensive PacBio long-read sequencing present a comprehensive view of fruit ripening-related AS, APA and lncRNA, which provides a better understanding of the complex post-transcriptional regulatory networks of fruit ripening in watermelon. Knowledge of the gene function and regulatory networks can make a significant contribution to the development of modern varieties that are able to withstand and adapt to the climatic conditions brought about by climate change.

## 4. Materials and Methods

### 4.1. Plant Materials

Watermelon *C*. *lanatus* (Thunb.) Matsum. & Nakai subsp. *vulgaris* cv 97,103 was used in this study. In order to make the fruit grow uniformly, we kept only one watermelon per plant. Flowers were hand-pollinated at 3–5 nodes and tagged. PacBio long-read sequencing was performed on three biological replicates of watermelon center flesh samples; these were collected at four developmental stages (10, 18, 26 and 34 DAP (days after pollination), Figure 1), each replicate resulting from the pooling of at least 10 fruits. Samples were rapidly frozen in liquid nitrogen and stored at −80 °C. All the samples used in this study were the same as from our previous study [30].

### 4.2. Total RNA Extraction, Construction of PacBio Library and Sequencing

Total RNA was isolated with a Total RNA Rapid Extraction Kit (Huayueyang Biotechnologies Co. Ltd., Beijing, China). Three biological replicates were carried out for Iso-Seq. The purity and concentration of total RNA were assessed using NanoDrop 2000, and the RNA integrity was evaluated with the Agilent 2100/4200 system. A total of 2 μg RNA was used for library construction. PacBio single-molecule real-time (SMRT) bell library preparation was performed using SMRTbell Express Template Prep Kit 2.0 (Pacific Biosciences, PN 101-853-100) in accordance with the manufacturer’s instructions. In brief, first strand cDNA was synthesized by reverse transcription after adding primer with Oligo dT paired with poly(A). Template switching was performed by adding the other primer paired with TS oligo. Full-length cDNA was obtained by cDNA amplification with barcoded cDNA PCR primers. Amplified cDNA was purified ProNex Beads and quantified by Qubit fluorometer (Invitrogen, Carlsbad, CA, USA). SMRTbell library was prepared using the Kit 2.0. Single-strand overhangs were removed, and then we performed DNA damage reparation, end-repair, A-tailing and adapter ligation. After library purification using 0.45X AMPure^®^ PB Beads and 1–10 Kb size selection, the library was prepared for sequencing for 30 h movie on the Sequel II system (Pacific Biosciences, Menlo Park, CA, USA).

### 4.3. Data Analysis

Watermelon reference genome and annotated gene models (97103, V2.5 version) were downloaded from CuGenDB (http://cucurbitgenomics.org/, accessed on 9 October 2023). The generated raw data were processed using the SMRT Analysis software (ISOseq version3.0). HiFi reads were generated by SMRTLink 10.0 software with parameters --min-passes = 3 --min-rq = 0.99. According to whether the sequence contained 5′ primers, 3′ primer or the poly(A) tail, the HiFi reads were divided into full-length and non-full-length sequences. Only the HiFi reads without any inter primer were considered as a full-length non-chimeric (FLNC) read. The FLNC were corrected by RNA-seq reads via LSC 1.alpha with Bowtie 2 v.2.2.1 alignment [51]. The error-corrected FLNC reads were then mapped to watermelon reference genome using minimap2 by the command (minimap2 -ax splice uf k14--sam-hit-only secondary=no) [31]. Alignment files were sorted, indexed, and filtered (-F 2048) by SAMtools. Stringtie2 (V2.2.1) was used for transcript assembly with the option -L –conservative [52]. Next, the collapsed transcripts were compared to reference annotation by Gffcompare (V0.12.6) with default parameter. MEME (V5.5.3) was used to search the upstream and downstream cis elements of polyadenylation sites [53]. KEGG pathway enrichment analyses were performed by the ClusterProfiler R- package, and the ggplot2 R-package was used for visualization [54].

### 4.4. Quantification of Differential Alternative Splicing

The AS event was identified by SUPPA2 (V2.3) generateEvents with options ‘-e SE SS MX RI FL -f ioe’ [55]. The quantification of AS event was performed by Salmon (V1.10.1) with options ‘—gcBias –validateMappings’ using RNA-seq. The differential alternative splicing events were quantified by SUPPA2 diffSplice using the option --method (empirical). Throughout, AS events with *p* < 0.05 were regarded as differential AS events.

### 4.5. Quantification of Differential Alternative Polyadenylation

Poly(A) site with >5% usage rate was remained as candidate poly(A) site. The distance between two distinct poly(A) sites should be more than 30 nt. The reads with termination site located 12 nt immediately upstream and downstream a poly(A) site was assigned to the poly(A) site. The poly(A) site with higher abundance was selected as the major sites. The abundance for each poly(A) site was normalized based on read count from the highest sites and library size. Fisher’s exact test was used for differential APA sites among the DAP10 vs. DAP18 group, DAP18 vs. DAP26 group and DAP26 vs. DAP34 group (*p* < 0.05).

### 4.6. Identification of lncRNA

For lncRNA identification, FLNC reads were merged together and aligned to genome by minimap2 [31] to generate BAM files, which were assembled to the transcripts by Stringtie2 [52]. Transcripts with only one exon and exon lengths of <200 bp were filtered out from downstream analysis. The prediction of lncRNA were preformed by CPC2, PLEK, lncFinder and Pfam with default options. The expression of lncRNA was quantificatied by Salmon with RNA-seq data.

### 4.7. PCR Validation of AS and APA

Two micrograms of total RNA was subjected to first-strand cDNA synthesis using a TransScript First-Strand cDNA Synthesis SuperMix (TransGen Biotech, Cat# AT301-02, Beijing, China) and an oligo (dT18) primer. AS was validated by RT-PCR using 2 × Phanta Max Master Mix (Dye Plus) (Vazyme, Cat# P525-01, Nanjing, China) with 5 × diluted cDNA as the template. The PCR products were visualized in 1.2% agarose gel stained by GelStain. The APA events were validated using a 3′ RACE Kit (Clontech, cat # 6106) according to the manufacturer’s instructions. Primer lists used in the validation are listed in Appendix A.

### 4.8. Data Presentation

All the statistical analyses and plots were carried out using Python matplotlib packages (version 3.8.0).

## Figures and Tables

**Figure 1 ijms-24-15333-f001:**
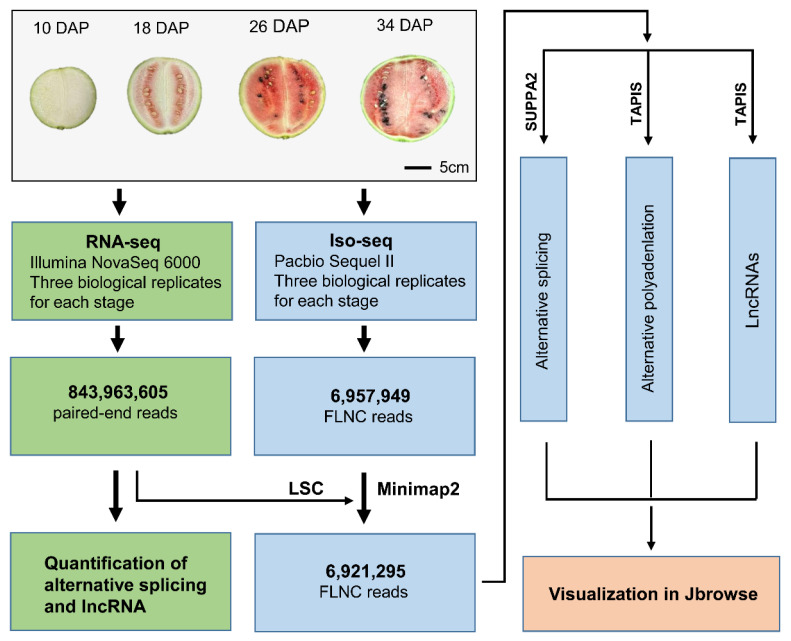
Flowchart of the experimental design and analysis for PacBio long-read sequencing and RNA-seq. In total, 6,921,295 mapped full-length non-chimeric (FLNC) reads were used for the downstream analysis, including alternative splicing (AS), alternative polyadenylation (APA) and lncRNA.

**Figure 2 ijms-24-15333-f002:**
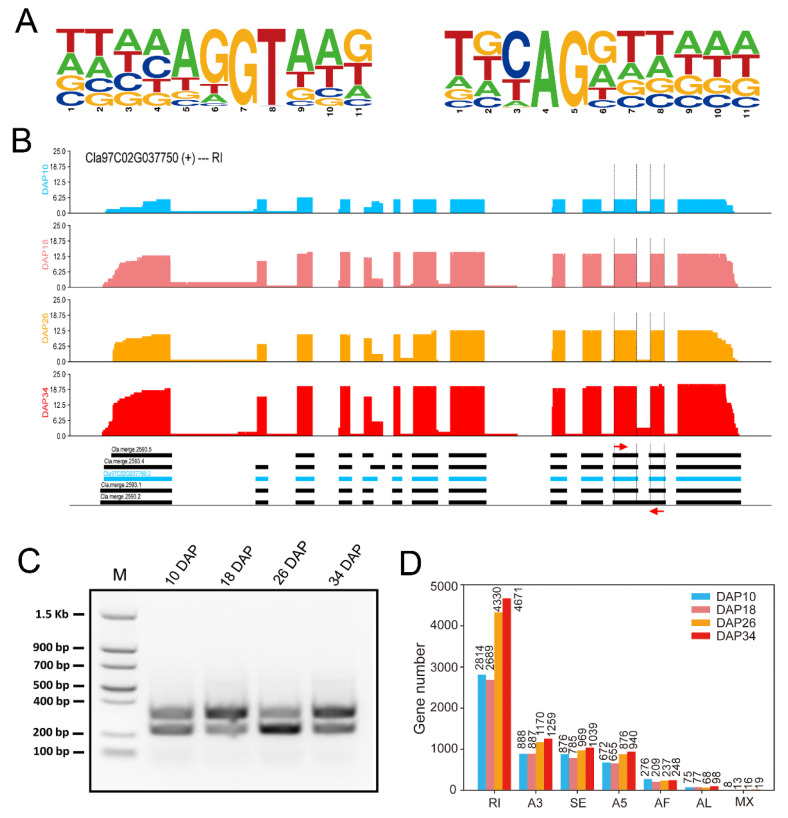
Comprehensive identification of splicing isoforms in watermelon fruit. (**A**) The nucleotide distributions flanking the 5′ donor (left) or 3′ acceptor sites (right). (**B**) Wiggle plot showing the intron retention event of the *PDS* gene (Cla97C02G037750). The red arrows represent the primers’ position. (**C**) Validation of the intron retention event in Figure 3B using RT-PCR. (**D**) Statistics of total AS events in watermelon fruit. RI: retained introns; SE: skipping exon; A5: alternative 5′ splice-site; A3: alternative 3′ splice-site; AF: alternative first exon; AL: alternative last exon; MX: mutually exclusive exons (MX).

**Figure 3 ijms-24-15333-f003:**
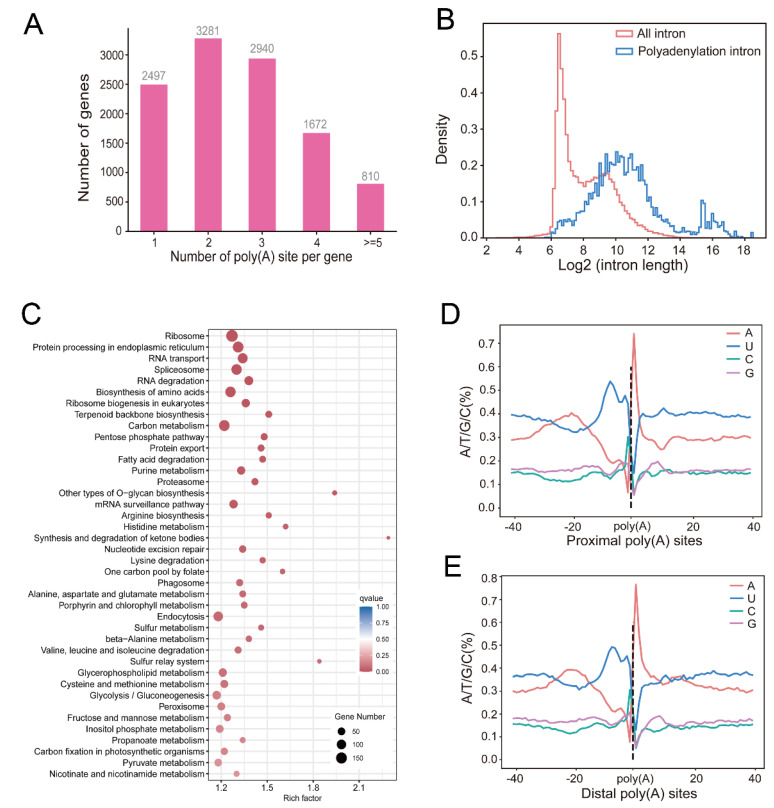
Genome-wide identification of alternative polyadenylation in watermelon fruit. (**A**) Distribution of the number of poly(A) sites per gene. (**B**) Length distribution introns with polyadenylation cleavage sites and all introns. (**C**) KEGG analysis of genes undergoing APA. (**D**,**E**) The nucleotide composition profile around proximal and distal poly(A) sites, respectively.

**Figure 4 ijms-24-15333-f004:**
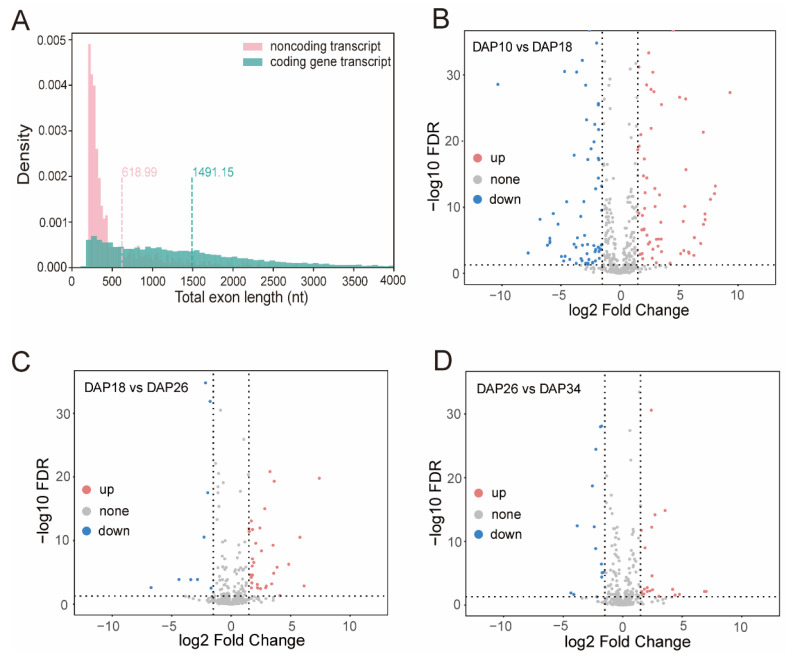
Identification of lncRNA in watermelon fruit. (**A**) The density distribution of exon length from lncRNA transcripts and protein-coding transcripts. Volcano plot showing differentially expressed lncRNAs: (**B**) in DAP10 vs. DAP18 group; (**C**) in DAP18 vs. DAP26 group; and (**D**) in DAP26 vs. DAP34 group.

**Figure 5 ijms-24-15333-f005:**
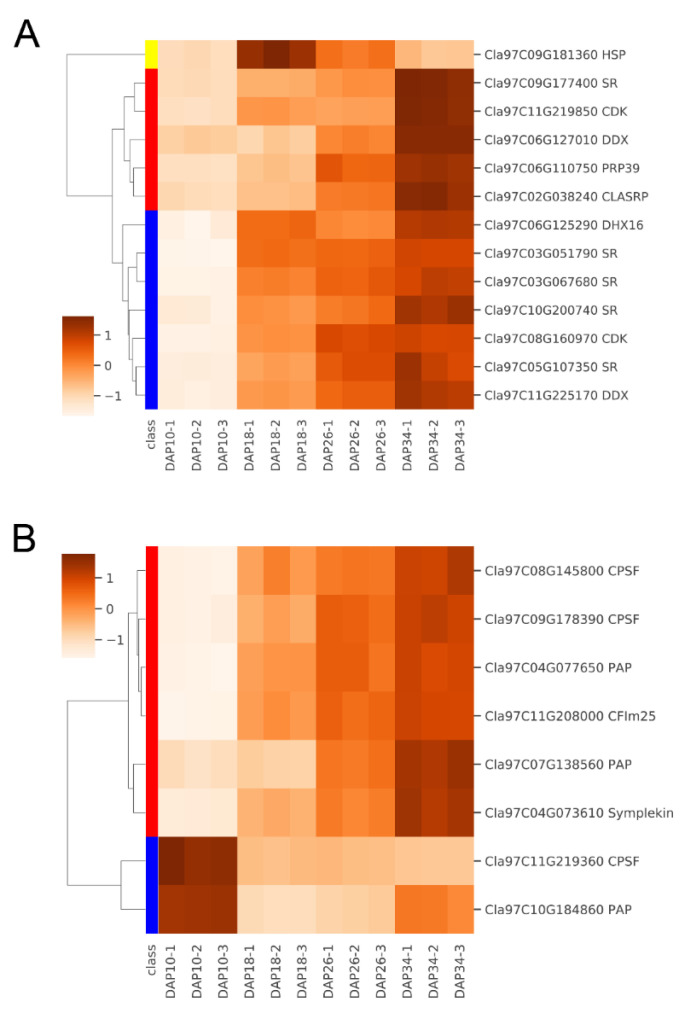
Heatmap of splicing factors and poly(A) factors in watermelon fruit. (**A**) Splicing-related factors. SR: serine/arginine-rich splicing factor; DDX: DEAD-box ATP-dependent RNA helicase; CDK: cyclin-dependent kinase; PRP39: pre-mRNA-processing factor 39; DHX16: DEAH-box protein, ATP-dependent RNA helicase; HSP: 35S_U5_associated_proteins; CLASRP: CLK4-associating serine/arginine rich protein. (**B**) APA-related factors. CPSF: cleavage and polyadenylation specificity factor; PAP: poly(A) polymerase; CFIm25: pre-mRNA cleavage factor Im 25 kDa subunit.

**Table 1 ijms-24-15333-t001:** Summary of PacBio sequencing.

Sample	cDNAsize	Number of Reads of Insert	Number of Five-Five Primer Reads	Number of Three-Three Primber Reads	Number of Adaptor Reads	Number of Filtered Short Reads	Number of Full-Length Non-Chimeric Reads	Error-Corrected and Mapped FLNC
DAP10-1	1-10k	632,678	19,259	7905	3	0	580,464	577,193
DAP10-2	1-10k	726,681	12,245	22,527	12	0	663,057	661,727
DAP10-3	1-10k	639,781	146,488	7345	18	0	463,837	460,828
DAP18-1	1-10k	727,467	20,973	7599	14	0	681,511	679,326
DAP18-2	1-10k	696,383	104,864	12,243	9	0	557,964	555,213
DAP18-3	1-10k	690,537	29,087	11,170	15	0	629,609	626,298
DAP26-1	1-10k	618,696	21,428	12,025	6	0	573,505	569,181
DAP26-2	1-10k	707,380	24,627	14,069	7	0	654,125	651,197
DAP26-3	1-10k	550,489	13,507	5581	7	0	520,269	516,792
DAP34-1	1-10k	587,052	98,357	18,328	42	0	457,391	453,471
DAP34-2	1-10k	531,201	11,605	5697	3	0	502,478	499,375
DAP34-3	1-10k	729,105	27,788	14,078	3	0	673,739	670,694

## Data Availability

Our PacBio Iso-seq data were submitted to the Sequence Read Archive (SRA) of NCBI under accession number PRJNA718122.

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
