# Peer review of "Comprehensive Profiling of Alternative Splicing and Alternative Polyadenylation during Fruit Ripening in Watermelon (Citrullus lanatus)"

_ijms, 2023, doi:10.3390/ijms242015333_

Round 1

Reviewer 1 Report

Watermelon, a cucurbitaceous plant, is an important crop grown worldwide. However, research on gene expression and its regulation during fruit ripening is undeniably lacking compared to other crops. The development of long-read sequencing technology has also made it possible to look at the diversity of splice variants and RNA modifications, which was not possible with short-read sequencing analysis. While data on model plants and major cultivated crops have been accumulating, primary data on watermelon are However, for watermelon, data are scarce.

The authors analyzed the transcriptome of watermelon fruit during the ripening process with PacBio and gave an overview of alternative splicing, alternative polyadenylation and long non-coding RNAs. These data will contribute to a molecular understanding of watermelon fruit ripening that will advance in the future and will be basis for future breeding research. The following are comments.

(1) In Materials and Methods, it is recommended that the sample preparation method be described (even if it is the same as in the previous study).

Author Response

Thanks for this suggestion. We have added the detailed sample preparation method  in the revised manuscript. And the added sample preparation method is "In order to make the fruit grow uniformly, we keep only one watermelon per plant. Flowers were hand-pollinated at 3-5 nodes and tagged. PacBio long-read sequencing were performed on three biological replicates of watermelon center flesh samples collected at four developmental stages (10,18, 26 and 34 DAP (days after pollination), Fig. 1), each replicate resulting from the pooling of at least 10 fruits. Samples were rapidly frozen in liquid nitrogen and stored at -80 °C."

Reviewer 2 Report

The manuscript addresses an important issue regarding profiling of polyadenylation and alternative spilicing in watermelon as  post-transcriptional regulatory mechanisms of fruit ripening and quality. The study is well structured, providing new insights of the molecular mechanisms underlying the complex regulatory networks of watermelon fruit ripening. A number of key genes implicated in fructose and mannose metabolism, starch and sucrose metabolism and carotenoid biosynthesis were found to undergo polyadenylation and alternative spilicing during watermelon ripening underscoring their role in the process for further research.

The authors provide a well-presented study that merits publication. A few minor issues that should be elaborated to increase the readership are as follows:

1.      In the conclusion section it would be nice to provide a closing statement regarding the application of the insights gained in fruit production quality and cost production to feed the expanding population in climate change.

2.      In legend of Fig. 2(D) it would be nice to briefly indicate the abbreviations as follows, RI: retained introns; SE: skipping exon; A5: alternative 5’ splice-site; A3: alternative 3’ splice-site AF: alternative first exon; AL: alternative last exon; MX: mutually exclusive exons (MX).

3.      Please correct a couple of grammar typo errors, such as:

a.      ln 81 “for above 12 samples” should be “for the above 12 samples.

b.      ln 251” fruit ripening-realted” should be “related”.  

English is fine except a couple of minor typo errors.

Author Response

QUESTION 1 of REVIEWER 2:

  1. In the conclusion section it would be nice to provide a closing statement regarding the application of the insights gained in fruit production quality and cost production to feed the expanding population in climate change.

ANSWER: Thank you for your advices. We have provided the closing statement in the revised manuscript. And the added content is “Climate change will have profound effects on food production. The production and quality of fresh fruit can be directly or indirectly affected by exposure to high temperatures and elevated levels of carbon dioxide and ozone. Accelerated crop improvement (traditional breeding methods, genetic modification and gene editing), new cultivation and processing techniques are required to produce sufficient food supplies and/or economize the production cost, which meet the basic nutrient requirements of the growing human population in climate change. Our comprehensive PacBio long-read sequencing present a comprehensive view of fruit ripening-related AS, APA and lncRNA, which provides a better understanding of the complex post-transcriptional regulatory networks of fruit ripening in watermelon. Knowledge of the gene function and regulatory networks can make a significant contribution to the development of modern varieties that are able to withstand and adapt to the climatic conditions brought about by climate change.”

QUESTION 2 of REVIEWER 2:

  1. In legend of Fig. 2(D) it would be nice to briefly indicate the abbreviations as follows, RI: retained introns; SE: skipping exon; A5: alternative 5’ splice-site; A3: alternative 3’ splice-site AF: alternative first exon; AL: alternative last exon; MX: mutually exclusive exons (MX).

ANSWER: Thank you for this suggestion. We have added the abbreviations in the legend of Fig. 2(D).

QUESTION 3 of REVIEWER 2:

  1. Please correct a couple of grammar typo errors, such as:
  2. ln 81 “for above 12 samples” should be “for the above 12 samples.
  3. ln 251” fruit ripening-realted” should be “related”.

ANSWER: Thank you for the reminder. We have corrected the two mistakes in the revised manuscript.

Reviewer 3 Report

The manuscript by Yu et al. describes the alternative splicing (AS) and alternative polyadenylation (APA) during ripening in watermelon. The authors perfomed PacBio single-molecule long-read sequencing technology to identify and compare the AS and APA events at four developmental stages during fruit maturation. The research was appropriate designed, the results were interesting, clearly presented and well discussed. In general, I believe that the manuscript was written in a comprehensive manner. Therefore I suggest that the paper should be accepted in the present form - the only correction needed is to place Fig.2 in the right position in the text. 

Author Response

Thank you for the reminder. This is a typesetting problem. We will ask the assistant editor to help us correct the mistake.